# The Relevance of Telomerase and Telomere-Associated Proteins in B-Acute Lymphoblastic Leukemia

**DOI:** 10.3390/genes14030691

**Published:** 2023-03-10

**Authors:** Tales Henrique Andrade da Mota, Ricardo Camargo, Estefânia Rodrigues Biojone, Ana Flávia Reis Guimarães, Fabio Pittella-Silva, Diêgo Madureira de Oliveira

**Affiliations:** 1Laboratory of Molecular Pathology of Cancer, University of Brasilia, Brasilia 70910-900, Brazil; 2Laboratory of Molecular Analysis, Faculty of Ceilândia, University of Brasilia, Brasilia 72220-275, Brazil; 3Brasília Children’s Hospital José Alencar, Brasilia 70684-831, Brazil

**Keywords:** *hTERT*, telomerase, telomere, leukemia, acute leukemia

## Abstract

Telomeres and telomerase are closely linked to uncontrolled cellular proliferation, immortalization and carcinogenesis. Telomerase has been largely studied in the context of cancer, including leukemias. Deregulation of human telomerase gene *hTERT* is a well-established step in leukemia development. B-acute lymphoblastic leukemia (B-ALL) recovery rates exceed 90% in children; however, the relapse rate is around 20% among treated patients, and 10% of these are still incurable. This review highlights the biological and clinical relevance of telomerase for B-ALL and the implications of its canonical and non-canonical action on signaling pathways in the context of disease and treatment. The physiological role of telomerase in lymphocytes makes the study of its biomarker potential a great challenge. Nevertheless, many works have demonstrated that high telomerase activity or *hTERT* expression, as well as short telomeres, correlate with poor prognosis in B-ALL. Telomerase and related proteins have been proven to be promising pharmacological targets. Likewise, combined therapy with telomerase inhibitors may turn out to be an alternative strategy for B-ALL.

## 1. Introduction

Leukemia is characterized by the production of abnormal leukocytes based on cytogenetic alterations, molecular modifications, clinical features and, notably, high proliferation [1]. Although knowledge on molecular alterations that lead to leukemogenesis as well as in the mechanisms involved in disease maintenance and propagation has gradually increased over the years, the identification of novel strategies to treat the disease is still lacking. Telomeres and telomerase are closely associated with cell proliferation, which makes them attractive targets for studies in oncology [2]. Telomerase is a reverse transcriptase that elongates the telomeres, thereby compensating the loss of telomere repeats after successive replication cycles; this is a phenomenon integrated to carcinogenesis known as cell immortalization [3]. Therefore, telomerase activity (TA) is detectable in almost all types of malignant cells, including leukemia cells. In this article, we discuss the relevance of telomerase and other telomere-associated factors to B-acute lymphoblastic leukemia (B-ALL) as well as their implications for development of new treatments, highlighting the role of these proteins as potential markers of B-ALL.

## 2. Materials and Methods

We conducted a literature search using the NCBI database (PubMed) in September 2022 using the following combinations of keywords: (“telomerase” OR “*hTERT*”) AND (“Acute lymphoblastic leukemia” OR “B-ALL” OR “ALL-B” OR “ALL”) and (“telomerase” OR “*hTERT*”) AND (“Acute lymphoblastic leukemia”) AND “treatment”. Additional references were obtained from cross-referencing bibliographies.

## 3. Discussion

### 3.1. Acute Lymphoblastic Leukemia

Acute leukemias are characterized by an uncontrolled proliferation of myeloid precursor cells or lymphoid precursor cells, with high rates of blasts in the blood as well as predominance of malignant cells [4]. Acute lymphoblastic leukemia (ALL) originates from B-cell precursor lineages (B-ALL) or, at a lower frequency, T-cell precursor lineages (T-ALL) [5]. Both share multiple subtypes of structural chromosomal alterations—aneuploidy, chromosomal rearrangements, and mutations—which are usually related to the development of B and T cells and cell cycle regulation [6].

The ALL is responsible for over 70% of the different types of leukemia that affect children [7]. The overall 5-year event-free survival rate for this disease exceeds 90% in developed countries [8]; however, 10–20% of patients succumb to recurrences, with high lethality [9]. Unlike pediatric ALL, adult ALL historically has a poor prognosis, with limited treatment options and a cure rate under 40% [10]. Adult patients have more cooperative mutations, which promotes epigenetic modifications that can lead to B-cell development [11]. The standard treatment for acute leukemia, both in adults and children, has been focused on high-intensity induction chemotherapy. In some cases, when chemotherapy is ineffective, hematopoietic cells can be transplanted to eradicate residual disease [12]. Nevertheless, cell transplantation is not recommended for all patients [13,14,15,16]. Despite treatments having increased survival rates in most cases, it is not uncommon for patients to develop resistance to treatment or to relapse over the years [17,18], making a constant search for new therapeutic strategies necessary.

The B-ALL cells and T-ALL cells have unique molecular markers. Some of them are based on chromosomal rearrangements and lymphocyte mutations, and others, such as cytokine receptors and protein kinases, are based on the dysregulation of signaling pathways [19]. These markers make up a cellular profile that is usually associated with prognostic or response to therapy. Nevertheless, the identification of new features is beneficial to improve the predictive potential of these molecular signatures.

### 3.2. B-ALL and Its Molecular Markers

B-ALL is the most common cancer in children [20]. According to the 2016 classification by the World Health Organization, there are more than 11 different B-ALL types. Among them, there are the old classifications BCR-ABL1 +ALL (Ph+), BCR-ABL1-like B-ALL (Ph-like or Ph-), and the *KMT2A* rearrangement, also known as MLL (mixed-lineage or myeloid-lymphoid leukemia), consisting of the rearrangement of the lysine methyltransferase 2A encoding gene to a highly diverse range of partner genes [21].

The Ph+ translocation t (9;22) (q34; q11) leads to a smaller chromosome [22]. It is less present in children, but its prevalence increases with aging, resulting in a greater occurrence in adults [23]. This translocation results in expression of the fusion gene BCR-ABL1 [24]. BCR-ABL encodes a tyrosine kinase protein that promotes uncontrolled proliferation and inhibition of apoptosis [25,26]. Ph+ clinical protocols have the main focus on multiagent chemotherapy in combination with tyrosine kinase inhibitors; this combination significantly improved outcomes in adults with newly diagnosed BCR/ABL1-positive ALL [27]. However, eventual relapses are inevitable [6].

The Ph-like type affects mainly male patients, and receives its name due to its gene expression profile, which is similar to Ph+, without the BCR-ABL1 fusion protein. The Ph-like ALL patients are up to 20% adolescents and young adults, but this disease can also be diagnosed during childhood; this occurs in approximately 3%, and is associated with worse outcomes [28,29]. This type of leukemia also shows complex genomic and genetic changes that deregulate cytokine receptors and tyrosine kinase proteins [30,31,32]. Around 50% of Ph-like cases exhibit overexpression of cytokine receptor-like factor 2 (*CRLF2*). However, mutations in other genes such as *IKZF1*, *ABL1*, *JAK2*, *ABL2*, *PDGFRB*, *TYK2*, *CSF1R*, *CRLF*, and *EPOR* may also be related to this modality of B-ALL [33]. The major modifications are the translocation of the JAK-STAT family and alterations in *CRLF2* [34,35,36].

The translocation t (4;11) (q21; q23) results in a genetic fusion between KMT2A-AF4, which is an alteration that mostly affects children with ALL. In addition, there are other known rearrangements involving *KMT2A*, such as KMT2A-AF10, KMT2A-AF9, and KMT2A-ENL [37,38,39]. These chromosomal rearrangements produce oncofusion proteins that harm the differentiation of hematopoietic stem cells [40]. The KMT2A-AFF1 translocations arise in utero and rapidly lead to the development of overt ALL. KMT2A-AF4 and AML1/MTG8 are associated with poor differentiation. This rearrangement is an important factor that supports *hTERT* expression due to a link between self-renewal and the transcriptional programs of leukemia cells [41].

In addition, a large number of new translocations serving as biomarkers in B-ALL have been identified, including rearrangements in *DUX4*, *ZNF384*, *MEF2D*, *MYC* and *NUTM1*. Mutation markers, such as *PAX5-P80R* and *IKZF1-N159Y*, are also used as biomarkers. Moreover, small telomere and telomerase activities have been reported as potential biomarkers in different oncological patients [42,43,44]. Furthermore, telomere maintenance and *hTERT* expression are being considered as potential targets in leukemia [45,46,47]. In this context, the relevance of using biomarkers to understand B-acute lymphoblastic leukemia, as well as for guiding therapies, is evident. The search for new markers is therefore essential to improve clinical conduct.

### 3.3. Telomeres, Shelterin Complex and Blood Cells

The telomeric tandem repeat sequences of TTAGGG are located in the end of chromosomes. They are bound by a specialized protein complex known as shelterin [48]. The telomeres play vital roles in cellular processes due to their capacity to protect chromosomes from end-to-end fusions and genome instability [49,50]. Cells with absent telomere maintenance mechanisms exhibit a maximum cell division capacity. Due to the loss of chromosome-capping function in telomeres, the cell enters senescence or is lead to apoptosis [51,52]. Furthermore, G-rich telomere repeat sequences are susceptible to oxidative damage, which reinforces telomere shortening and leads to cell senescence related to aging [53].

The shelterin complex is composed of six protein subunits—TRF1, TRF2, RAP1, TIN2, ACD, and POT1 (Figure 1) [54]. Although shelterin have many functions, such as protecting the telomeres from DNA deterioration and preventing activation of unwanted repair systems, they also play a key role in telomerase activity regulation [55,56,57].

Shelterin are also involved in the establishment of heterochromatin and telomeric silencing. The recruitment of these proteins is apparently related to enriching methyltransferases into the sub-telomere regions for gene silencing, which allows telomere lengthening [58].

The composition of white blood cells depends on different exposures to stress factors [59]. Different stressors can initiate a redistribution of leukocytes from immune reservoirs to the circulation. This is relevant due the fact that telomere length (TL) differs among leukocyte subtypes (lymphocytes, monocytes, granulocytes). Moreover, naïve leukocytes have telomeres similar to those found in hematopoietic stem cell progenitors, while smaller telomeres are present in mature leukocytes. This makes it difficult to define if alterations in TL can be attributed to a blood sample leukocyte composition or to a particular condition [60,61].

### 3.4. Telomerase and Cancer

The telomerase consists of the catalytic telomerase reverse transcriptase subunit, known as TERT, and an RNA component (hTR) that works as a template for telomere extension [62,63,64]. The canonical functions of telomerase are related to telomere length maintenance and genome stability, while the non-canonical functions are involved in the regulation of non-telomeric DNA, alterations in cell cycle kinetics, the rise of proliferation, chromatin remodeling and more (Appendix A) [65,66,67].

Telomerase acts on TL maintenance during the fetal phase of life, and its presence in adult tissues is infrequent. In leukocytes, TL is stabilized around age 20, and a slow rate of attrition occurs during adulthood [68]. Moreover, strong telomerase activity is found in progenitor stem cells and activated lymphocytes, and it is especially enhanced in carcinogenesis, with implications for genome integrity, proliferation and stemness [69].

Telomerase is present in tumor cells from over 85% of cancer types, while about 15% of them continue the telomere lengthening through homologous recombination processes collectively known as alternative lengthening of telomeres (ALT), which is not a telomerase-dependent mechanism [70,71,72].

The regulation of telomerase activity is crucial and occurs mainly through the control of *hTERT* transcription, which also determines in which type of cell telomerase will be expressed [62]. The regulation of the active enzyme, on the other hand, is performed by a post-transcriptional maturation process involving binding to hTR (which is constantly expressed) [48]. Moreover, the regions containing TERT and hTR genes, 5p15.33 and 3q26.3, respectively, are usually amplified in cancer cells [73]. However, the *hTERT* mutation itself has been shown to be insufficient for telomere maintenance [74].

Mutations in the *hTERT* promoter represent frequent somatic genetic alterations that cause telomerase reactivation [75,76]. Epigenetic changes are also involved in different steps of this reactivation, including DNA methylation of *hTERT* controllers that are associated with transcription activators such as c-*myc*, *MZF-2*, and *WT-1*. Hypermethylation prevents binding of the repressors to the promoter, which leads to *hTERT* upregulation and telomerase activation [77,78].

Telomerase was first described for its capacity to elongate telomeres [79]. Nevertheless, it is becoming clear that TERT is also involved in distinct biological pathways [80] that are related to both physiological and pathological processes. These processes include those that contain stem cell functions, homeostasis, aging, tumor progression, drug resistance, regulation of non-telomeric DNA damage responses, promotion of cell growth and proliferation, acceleration of the cell cycle and damage to mitochondrial DNA, which influences cell integrity following oxidative stress. For these non-canonical activities, telomerase was reported to act on the activation of the senescence signaling pathway, the induction of apoptosis through mitochondrial pathways, autophagy, cellular growth, *NF-kB* mediated inflammation, and cancer progression in general (Appendix A).

Telomerase expression and activity are also influenced by factors from distinct pathways. P23, for example, acts as an anti-apoptotic factor that plays a significant role in estrogen receptor α signal transduction, but which can also regulate TA by binding directly to the catalytic subunit of telomerase. This interaction is required for TL maintenance for an efficient telomerase assembly, helping to modulate telomerase–DNA binding in extension activities. Thus, the overexpression of *p23* causes B-ALL cells to evade apoptosis for both TERT-related and independent pathways [81]. Similarly, *c*-MYC promotes *hTERT* deregulation, resulting in the reduction of telomere length, telomerase activity and cell proliferation [82]. Additionally, TERT can act as a transcription co-factor that regulates expression of several genes [77], which are summarized in Appendix A [83,84,85,86,87,88,89,90,91,92,93,94,95,96,97,98,99,100,101,102,103,104,105,106,107,108,109,110,111,112,113,114,115,116,117,118,119,120,121,122,123,124,125,126,127,128,129,130,131,132,133,134,135,136,137,138,139,140,141,142,143,144,145,146,147,148,149,150,151,152,153,154,155,156,157,158,159,160,161,162,163,164,165].

Interestingly, there is evidence that TA is also related to gender. Male Egyptian B-ALL patients, for example, were reported to have higher expression and activity of TERT than female patients. In this particular study, the total leukocyte count of both groups of patients was higher when telomerase is upregulated, indicating a poor response to therapy [166].

The multi-functional profile of telomerase, as well as its relevance for carcinogenesis and cancer maintenance, make it an extremely relevant target for the development of studies focusing on both therapeutical purposes and on the understanding of tumor biology, especially in blood-related diseases [167,168,169].

### 3.5. Telomerses and Telomerase in B-Acute lymphoblastic leukemia

The *hTERT* mRNA can be detected in memory and naïve germinal center B-cells (GC) in which its level is associated with high TA [170,171]. The expression of telomerase in GC B-cells is inducible during immunological response, but at lower levels than in leukemic cells [172]. The synergistic stimulation with anti-IgM Ab plus specific cytokines (IL-2, IL-4, and IL-13), as well as the surface molecules BCR or CD40, increase telomerase activity in B-cells. Then, the canonical telomerase functions seem to be the main mechanism for telomere length maintenance in the germinal center in normal (non-pathological) conditions (Figure 2).

Despite the fact that up-regulation of telomerase in human B lymphocytes may occur independently of cellular proliferation, with expression of telomerase catalytic subunits [173], it has been demonstrated that telomerase activity can also be induced by PI-3 kinase-dependent and independent pathways linked to proliferation. For instance, the inhibition of PI3K blocked the anti-IgM plus anti-CD40-induced telomerase expression in B cells in a dose-dependent manner [172,174].

Telomerase is virtually absent in most adult tissues and detectable in most tumors, but the physiological role of this enzyme in lymphocytes represents an important challenge for approaching it as biological marker in leukemia. However, the straight relationship between telomerase activity and proliferation, as well as its anti-apoptotic role [175], make it essential for leukemogenesis. The uncontrolled proliferation of B lymphoblastic precursor cells in B-ALL leads to shortened telomeres and raised telomerase activity [176,177]. Additionally, leukemia cells with a normal karyotype exhibit longer telomeres when compared with cells with abnormal karyotypes [176,177].

Both high telomerase activity and shortened telomeres are correlated with disease progression, resistance to therapy and bad prognosis in ALL [178]. Telomerase can block apoptosis mechanisms in leukemic blasts, resulting in faster disease progression, and its activities are related with lactate dehydrogenase, which is an unfavorable prognostic factor for ALL patients [166]. In this sense, recent studies have revealed telomerase overexpression and *hTERT* methylation status as a promising prognostic biomarkers in B-ALL (especially for childhood disease), and more precisely for maintenance and disease persistence, also reinforcing the potential of telomerase as therapeutical target, mainly due to its multiple non-canonical actions [45,47,179].

In Ph+ B-ALL, the p16INK4A/pRb pathway with a high TA determines a group of adult ALL associated with poor prognosis [180]. Furthermore, Philadelphia chromosome genes also regulate telomerase and its activity at multiple levels [181]. Antisense Inhibition of BCR/ABL, for example, is able to enhance telomerase activity, leading to activation of tyrosine kinase proteins and inhibition of apoptosis [25,26].

However, controversial results can be found in the literature. In the work of Ozgur et al. [182] and Eskandari et al. [179], no significant association was found between *hTERT* mRNA expression and hematological parameters in B-ALL. Nevertheless, these same studies have showed that telomere attrition is linked to childhood ALL. On the other hand, Borssén et al. have demonstrated that B-cell precursor group cases had a higher *hTERT* methylation than diploid ALL. In addition, *hTERT* mRNA levels were negatively associated with methylation status, but curiously, in low-risk B-cell precursor patients, long telomeres indicated a worse prognosis [183].

Monitoring minimal residual disease (MRD) is one of the most important strategies to follow up B-ALL patients due the capacity to identify lower cell levels. In this sense, it was demonstrated that quantification of telomerase expression along with monitoring MRD by qPCR can strengthen the follow up of patients with B-ALL. This would not only improve treatment follow up but also help to identify post-therapy remission [47].

Finally, a large number of studies propose that pathogenesis and the phenotypic characteristics of B-acute lymphoblastic leukemia are connected with the conjunction of specific targets and DNA variations promoted by epigenetic alterations such as methylation [184,185,186]. *hTERT* promoter methylation is infrequent in B-ALL cases with remission, and there is no association with TL. However, *hTERT* RNA expression is reduced when methylation occurs [183]. Methylation of CDKN2B CpG island was associated with high telomerase activity in children with B-ALL [187]. It has also been shown that β-Arrestin1 promotes cellular senescence in B-ALL by binding with *P300-Sp1* in order to regulate *hTERT* transcription. In that case, *hTERT* is a major factor due to the regulation stimulated on the *β-Arrestin* pathway, rising *p300-sp1* expression [87].

### 3.6. Shelterin in B Lymphoblastic Leukemia

The role of the shelterin complex in B-ALL has also been studied. *TRF2* expression was shown to be increasing in acute leukemias and also higher in lymphocytes of B-ALL patients, particularly in those with an abnormal karyotype [177]. Recently, NOTCH3, PAX5, CBFB, and particularly ACD were shown to drive the activated RAS pathway and monosomy 7 to B-acute lymphoblastic leukemia [188]. Nonetheless, ACD plays a key role in telomere maintenance due to its interaction with POT1; this combination protects telomeres and recruit telomerase at chromosome ends. Despite the overexpression of wild-type, ACD does not lead to telomere lengthening, the G223V mutation reflects on TL and seems to be related to decreased apoptosis activity in B-ALL cells, that is triggered to the functional role of ACD and its relevance for cell survival in leukemia [189].

Beyond hTERT, B-ALL patients also show high expression of *CTC1* and *OBFC1* (they are part of CST complex which works with the shelterin complex to lengthen telomeres); however, only *CTC1* was associated with leukemia [190].

### 3.7. Telomerase and Genetic Variation

There is, currently, a vast field literature on *hTERT* polymorphisms and their implications in oncology, but just few works approach it in the context of B-ALL. The *hTERT* polymorphisms rs2735940 and rs2736100, for example, were defined as risk factor for ALL and turned out to be functional; they were implicated in TA, TL and homeostasis. The same authors showed that a variant near hTR, as well as high TL, are markers for risk of acute lymphoblastic leukemia in Chinese children [191].

Another study demonstrated that the survival rate of children with B-ALL was higher in European American children (EA) than in African American children (AA), which appeared to be due to the different canonical pathways affected in each case. Telomerase signaling is related to AA pathways, while chromosome aberrations in EA more frequently affect genes involved with homologous recombination [192]. This suggests that *hTERT* may have a different influence on B-ALL with regard to different populations; nonetheless, large-scale studies need to be done to verify this hypothesis.

### 3.8. Current Telomerase Inhibitors and Their Clinical Potential

Different approaches for telomerase inhibition have been under development for more than a decade, aiming at more effective treatment strategies. Telomerase activity can be inhibited by different strategies, such as disrupting biosynthesis, maturation, assembly, or correct interaction between the telomerase complex and the substrate [193]. In Table 1, we exemplified some of the currently available telomerase inhibitors.

Doxorubicin (DOX) is a chemotherapy drug used in different cancer treatment protocols. This molecule promotes cell death through disruption of DNA repair by inhibiting topoisomerase II, and provokes oxidative stress by generating free radicals [7,206]. However, doxorubicin has a high toxicity for the heart, which can lead to mortality among cancer patients, limiting its clinical applications [207].

The *Zataria multiflora* extract (ZME) is a plant extract oil that exhibited a synergistic effect in association with doxorubicin, increasing its toxicity in all tested B-ALL cell lines. Despite this combination raising the levels of anti-apoptotic Bcl-2, it downregulated expression of *c-Myc* and *hTERT*, showing ZME as a potential adjuvant for treatment of pre-B-acute lymphoblastic leukemia [198]. Additionally, the telomerase inhibitor MST-312 decreased in vitro effective dose of doxorubicin. The combination MST-312/DOX reduced cell growth and promoted apoptosis in -B-ALL cells through unbalancing the *Bax*/*Bcl-2* ratio aligned to down-regulation of *c-Myc* and *hTERT* [208]. It is important to mention that most TERT inhibitors are developed, aiming canonical function of telomerase, but there is evidence of the antitumor effect of MST-312 associated with non-canonical ones [207].

BIBR1532 is one of the most powerful telomerase inhibitors. This synthetic non-nucleoside compound binds to telomerase and acts as a chain terminator during nucleotide polymerization, inhibiting TA in a dose-dependent way [209,210]. BIBR1532 provoked cell death in pre-B-ALL cells after suppression of *hTERT* and *c-Myc* expression. Besides, high doses of BIBR1532 can induce *p73*, up-regulate *Bax* and activate caspase-3 [46]. Association of BIBR1532 with doxorubicin also reduced surviving expression and produced a synergistic anticancer effect in B-ALL through induction of ROS, which increased expression of *Bax*. Furthermore, it raised *p21* levels, which promoted G1 cell cycle arrest and downregulation of *p73*-mediated *c-Myc* and *hTERT* expression [211].

To summarise, the combination of DOX with BIBR1532, ZME or MST-312 increases its therapeutic effect. A synergistic mechanism, if confirmed, could lead to therapeutical protocols with a lower dose of doxorubicin, thereby decreasing the risk of DOX-induced cardiotoxicity.

It is important to mention that there are a vast number of works showing the antitumor effects of telomerase inhibitors in a variety of in vitro and in vivo models of different cancer types [212], including some with which clinical trials are in progress or already concluded [213]. There are also studies testing telomerase-targeted immunotherapy [214] and other telomere-related therapeutical strategies. In this review, we attempted to summarize information regarding telomerase in B-ALL, which is still very scarce. Nevertheless, any one of these prototypes, once proved effective for cancer control, has the potential to be used in the context of leukemia.

In any case, further investigation is deeply required, including clinical trials, which could determine the safety of these compounds alone or as combined therapies for B-ALL patients. However, the greatest challenge that must be overcome in order to take studies with telomerase inhibition to the next level is the development of compounds targeting inhibition of non-canonical functions, which have been demonstrated to be crucial for cancer maintenance.

## 4. Conclusions

The physiological functions of telomerase in lymphocytes represent a challenge to determine the role of this enzyme in B-ALL. However, clearly reviewed data showed evidence of the potential of telomerase and other telomere-related proteins as clinical biomarkers and pharmacological targets. Briefly, high telomerase activity or *hTERT* expression, as well as short lymphocytes’ telomeres, are frequently correlated with poor prognosis or even higher risk for B-ALL. However, there are still some apparent conflicting data in the literature, associating long telomeres with worse prognosis. We emphasize the word “apparent” since it also became clear that there is no single pattern concerning telomere and telomerase functions in leukemia, especially considering all possibilities of non-canonical actions of TERT. Additionally, the influence of telomerase on B-ALL seems to be divergent in different ethnic groups, which needs further investigation to be better elucidated. Finally, this pool of results shows a promising future for telomere and telomerase targeted therapy as new or combined treatments, but most data are too preliminary for short-term clinical use, especially in ALL.

## Figures and Tables

**Figure 1 genes-14-00691-f001:**
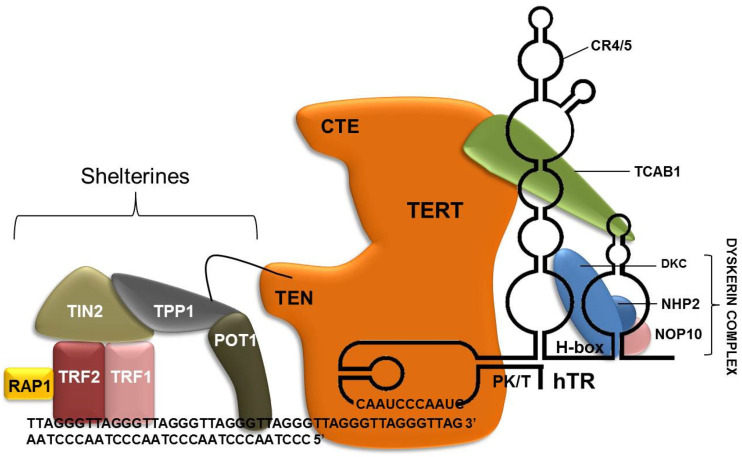
Telomerase representation. Dyskerin complex (NHP2, NOP10 and DKC) binds to hTR through its ACA domain. TCAB1 binds to TERT and to hTR. The template region of hTR binds to the telomeric 3′ -end strand. Shelterin binds to telomeric repeat region. TRF2 interacts with RAP1 and TRF1, binding directly to the telomeric DNA and to TIN2, which also binds to TPP1. TPP1 interacts with POT1, which is responsible for recruiting telomerase to telomeres through the TEN domain of TERT.

**Figure 2 genes-14-00691-f002:**
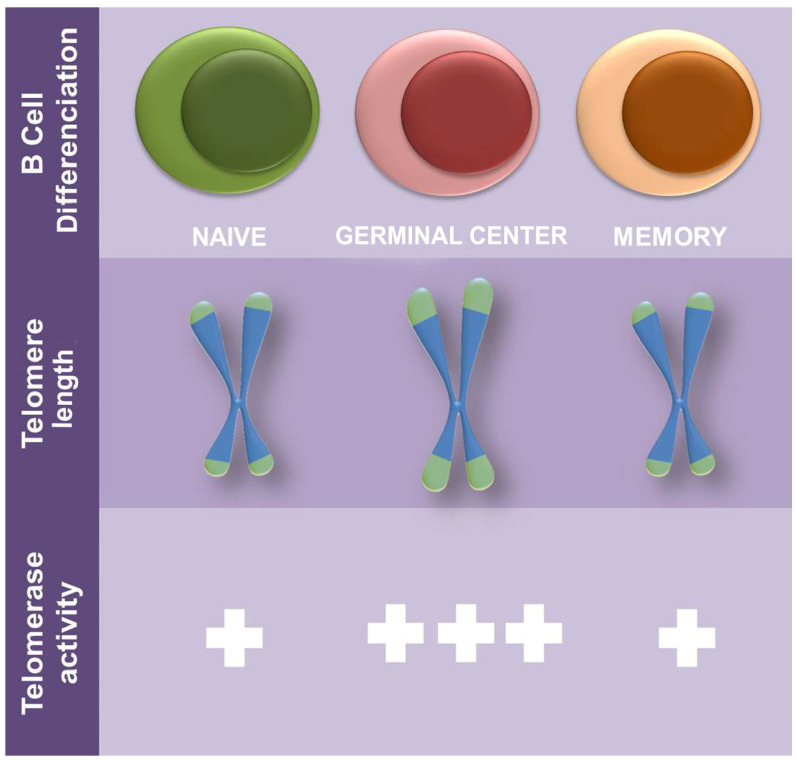
Model of telomere regulation in B cells. The green color represents telomere length and the blue color the chromosomes. The germinal center has greater telomere length than naive and memory cells.

**Table 1 genes-14-00691-t001:** Telomerase inhibitors.

Compounds	Chemical Structures	Type of Therapy	Main Findings	References
N,N′-1,3-Phenylenebis-[2,3-dihydroxy-benzamide] (MST-312)	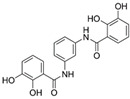	Combined with doxorubicin	MST-312 inhibits the progress of multiple myeloma by inhibiting the telomerase activity of this cells. Monotherapy long-term exposure to the MST-312 in U251 cells resulted in the induction of cell adaptations with possible negative clinical implications. MST-312 alters telomere dynamics, gene expression profiles and growth in human breast cancer cells	[194,195,196]
Zataria multiflora extract (ZME)	Chemical structures of main volatile and non-volatile constituents are in Sajed, Sahebkar and Iranshahi works [197]	Combined with doxorubicin	Pulmonoprotective action of Zataria multiflora ethanolic extract on cyclophosphamide-induced oxidative lung toxicity in mice Anti-leukemic effect of Zataria multiflora extract in combination with doxorubicin to combat acute lymphoblastic leukemia cells.	[198,199]
2-[(E)-3-naphtalen-2-yl-but-2-enoylamino]-benzoic acid (BIBR1532)	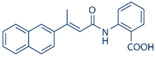	Monotherapy and combined with doxorubicin	BIBR1532 exerts a series of anti-cancer activities linked to the inhibition of the canonical telomerase pathway and the TERT extra-telomeric functions in feline oral squamous cell carcinoma. BIBR1532 exhibits a selective cytotoxicity against primary leukemia cells from acute myeloid leukemia and chronic lymphocytic leukemia patients. Telomerase inhibition by BIBR1532 causes rapid cell death in pre-B-acute lymphoblastic leukemia cells BIBR1532 exerted potent cytotoxic effects on a panel of human cancer cells in a dose-dependent manner in leukemic cells which were more sensitive to the inhibitor BIBR 1532, exerts a direct short-term growth suppressive effect in a concentration-dependent manner possibly through the downregulation of *c-Myc* and hTERT expression	[46,200,201,202,203]
lipid-conjugated N30-P50 thiophosphoramidate GRN163L (Imetelstat)	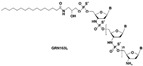	Monotherapy	Imetelstat induces leukemia stem cell death in pediatric acute myeloid leukemia. The telomerase antagonist imetelstat efficiently targets glioblastoma tumor-initiating cells leading to decreased proliferation and tumor growth The inhibition of telomerase with imetelstat ex vivo led to significant dose-dependent apoptosis of B-ALL cells. Thus, imeteostat can be usefull in the standard treatment of B-ALL	[45,204,205]

## Data Availability

Not applicable.

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
