# Peer review of "The Relevance of Telomerase and Telomere-Associated Proteins in B-Acute Lymphoblastic Leukemia"

_genes, 2023, doi:10.3390/genes14030691_

Round 1
Reviewer 1 Report
Dear Editor,
The authors have investigated the contributory role of telomerase in acute lymphoblastic leukemia (ALL). At first the authors have taken a glance at B-ALL and its molecular markers. Then, they continued with an overview of telomeres, telomerase, and shelterine complex. Finally, the authors have reviewed telomerase and shelterine complex in B-ALL.
It is a well-written review in the era of telomerase in ALL. However, there are minor comments which authors should consider.
Comments to authors:
1. The authors should summarize the main findings of the contributory role of telomerase in ALL and those studies which investigated telomerase inhibition, either as monotherapy or in a combined-modal strategy, in this leukemia as a Table.
2. It would be better if authors digest the results of previous studies, and make their own conclusion in the Abstract and Conclusion regarding: whether telomerase inhibition is a beneficial approach in B-ALL according to their viewpoint.
3. Moreover, it is recommended the authors provide a section concerning plausible challenges of telomerase inhibition in ALL.
4. Recommended References: There are several reports, in particular studding the effect of telomerase inhibition using BIBR 1532 in different leukemia other thea ALL, which may be integrated in Table 1.
· doi: 10.1016/j.ejphar.2019.01.018. which has investigated the contributory role of microRNAs in anti-cancer effects of BIBR1532 on promyelocytic leukemia cell line.
· doi: 10.3109/07357907.2011.629378. & doi: 10.3109/10428194.2012.704034. have investigated telomere –independent mechanism of action of BIBR1532 in APL cells either alone or in combination with chemotherapy.
5. It would be beneficial for readers of the genes if the authors could provide a schematic representation of their review.
Author Response
Dear Reviewer,
The authors have investigated the contributory role of telomerase in acute lymphoblastic leukemia (ALL). At first the authors have taken a glance at B-ALL and its molecular markers. Then, they continued with an overview of telomeres, telomerase, and shelterine complex. Finally, the authors have reviewed telomerase and shelterine complex in B-ALL.
It is a well-written review in the era of telomerase in ALL. However, there are minor comments which authors should consider.
We appreciate the comments.
Comments to authors:
1.
The authors should summarize the main findings of the contributory role of telomerase in ALL and those studies which investigated telomerase inhibition, either as monotherapy or in a combined-modal strategy, in this leukemia as a Table.
The main findings are now summarized in the new version of the conclusion. Putting this information into a new table, as suggested, would make it redundant, but we also updated the table 1 to add the type of therapy (monotherapy / combined).
- It would be better if authors digest the results of previous studies, and make their own conclusion in the Abstract and Conclusion regarding: whether telomerase inhibition is a beneficial approach in B-ALL according to their viewpoint.
We rewrote abstract and conclusion, as suggested.
- Moreover, it is recommended the authors provide a section concerning plausible challenges of telomerase inhibition in ALL.
We updated the section about telomerase inhibitors (now it is titled “current telomerase inhibitors and their clinical potential”), but it is important to mention that the focus of the article is discussing telomerase as a marker, we approached its potential as pharmacological target because it is a natural consequence of the role of the enzyme in cancer maintenance. In fact, there are many works showing experimental data on antitelomerase therapy in ALL, but we suppose that it is subject for another review.
- Recommended References: There are several reports, in particular studding the effect of telomerase inhibition using BIBR 1532 in different leukemia other thea ALL, which may be integrated in Table 1.
- doi: 10.1016/j.ejphar.2019.01.018. which has investigated the contributory role of microRNAs in anti-cancer effects of BIBR1532 on promyelocytic leukemia cell line.
- doi: 10.3109/07357907.2011.629378. & doi: 10.3109/10428194.2012.704034. have investigated telomere –independent mechanism of action of BIBR1532 in APL cells either alone or in combination with chemotherapy.
The recommended references were added.
- It would be beneficial for readers of the genes if the authors could provide a schematic representation of their review.
We submitted a schematic representation of the review as graphical abstract.
Reviewer 2 Report
The manuscript “The relevance of telomerase in B lymphoblastic leukemia” submitted to Genes is a fairly comprehensive paper with lots of information. However, frequent typos, grammatical errors and the lack of clear organization makes it difficult read through properly. There are also lots of inconsistencies in acronym usage (e.g. B-ALL vs ALL). There are also mentions of shelterin and other accessory proteins, but they aren’t technically part of telomerase. If they are to be mentioned, perhaps the title should be expanded to “relevance of telomere and telomerase in B lymphoblastic leukemia.” Some different sections also have different acronyms and style and redundancy of information, which makes it feel like they are written by different people but not integrated well together enough.
When introducing telomerase functions, there should be an explanation on the distinction between canonical and noncanonical functions. The only mentions of “non-canonical” are in the abstract and conclusion. This leaves the reader guessing whether they are reading about canonical or non-canonical functions. I suggest separating them into different sections: one talking about telomere length regulation in normal blood cells and B-ALL aspects and one about gene-level regulation of TERT on other signaling pathways. It is sometimes hard to differentiate the two, and in some cases, both might be involved at the same time. That is also fair and should be mentioned as a possibility and an important consideration when discussing etiology of the disease.
Another important point is that the Shelterin complex is defined set of proteins associated with the telomere. In some parts, it seems to be inferred that shelterin includes all the other protein partners related to telomerase that are not directly in the holoenzyme, which is incorrect. For subsection headers, perhaps the title of “Shelterin and other telomere-related partners” could be used.
Overall, I do like some of the information included (e.g. Table 1 and supplementary table 1) and there are definitely a lot of information in this manuscript. The tables actually have pretty nice information that is not even reflected in the text! The text itself is written in a way that is not efficient to read for readers. Please refer to the specific points below on some key issues identified. As can be seen from the list, there are also very many editorial issues that needs fixing (and likely more that I have not listed). Careful editing of the manuscript prior to next review would be greatly appreciated. Despite the organization/flow and editorial issues, I think this manuscript does have its merit but needs some major revisions.
Specific points:
Line 83-84: It is not immediately obvious how antisense of these genes affects telomerase activity. Might be more appropriate in a later section focusing on telomerase? Or some more details might be needed.
-Line 87: “with” needs to be removed. Otherwise, it reads like the quantification of these outcomes was to be included
-Line 105: Do not italicize MLL as it is not an official gene symbol under HGNC. The official gene symbol as you have mentioned is KMT2A
-Line 114: suggest “tandem repeat sequences”
-Line 117: please specific what instability; I think you meant “genome instability” here
-Line 119: Citation 50 is not the appropriate citation. That article is genome sequencing of TERT-positive vs ALT cancer. Given that telomere shortening is a well-accepted knowledge in the field, a review paper should suffice. Also note that the end replication problem is not the only explanation (might even be a minor one). T-loop resolution and DNA damage (like the oxidative damage mentioned right after) also contributes to shortening.
-Line 140: Why “exceptional”? Is that high or low? Since that word usually means very good. Please reconsider a more appropriate word. Perhaps you meant irregular or infrequent.
-Line 142-143: Remove “also possible to state.” Seems like low confidence statement and a slight overgeneralization of cancer. More accurate to say “Telomerase is upregulated in most cancer…” which is well known already now.
-Line 145: Suggest removing “Telomerase, a ribonucleic….ending replication obstacle.” It is repetitive so it’s unnecessary here.
-Line 147: It needs to be noted somewhere that ALT is not a telomerase-dependent mechanism.
-Line 149-151: This might be better placed in the start of the next paragraph. Here, it should be noted that TERT is the limiting factor since it is heavily regulated whereas TERC is ubiquitously expressed.
-Line 152-183: This section needs to be organized better for readability and flow (perhaps separating into to subsections or paragraphs would help). For example. 1) Canonical and non-canonical functions of TERT. 2) TERT mutation has been found in many cancers. 3) How cancer achieve immortalization through telomerase reactivation and TERT expression upregulation.
-Line 162-165: Are the two really distinct mechanisms? This should be a two-step mechanism. First phase, heal shortest telomere. Second phase, critical level short telomere leads to genome instability which drives telomerase activity up.
-Line 184: Wrong spelling for Shelterin. Please correct the rest. Shelterin is not typically referred in plural form.
-Line 184-210: This Shelterin part is strangely written without much aim. It is also very short so it doesn’t really convey much of anything. Since telomerase is the focus of this paper, perhaps shelterin may not need much mention other than an introduction. I suggest including this with earlier introduction to create an introduction to telomere.
-Line 228-240: Why is the CLL being discussed in the section for ALL? Furthermore, CLL has not even been introduced before. Given that the paper is focusing on ALL, perhaps CLL should be mentioned in the introduction but not be discussed any further later.
-Line 282: What’s TA? Please define.
-Line 281-288: I am unaware that p23 being a shelterin protein, so it is strange to be discussed under this section. P23 does associate with TERT though.
-Line 311: Aren’t there four compounds in table 1?
-Telomerase inhibitors section: Are any of these in clinical trials with known outcomes? This information would be useful to know.
-References: There are 214 references but in text there are only up to 124 citations. Where are the rest cited? Is that combined with supplementary?
Minor points (typos, grammatical, editorial errors):
-Line 12: grammatical error, suggest splitting to two sentences. “…carcinogenesis. Thus, telomerase has been largely studied in the context…”
-Line 13 and 19: suggest remove “in fact” as they are just filler words. Please remove these from the rest of the manuscript
-Line 24: “based in” should be “based on”
-Line 27: suggest change to “a reverse transcriptase that elongates the telomere and thereby compensating the loss…”
-Line 35-40: Is Materials and Methods appropriate or even necessary for a review article?
-Line 59: should be “aiming at”
-Line 62: “have” should be “having”
-Line 66: “unique” is a better term than “particular” here
-Line 69: “cell” should be “cellular”
-Line 70: suggest changing “welcome” to something else such as “beneficial”
-Line 79: suggest changing “outturns to” to “results in”
-Line 86: combination “with”
-Line 95: dash should be a comma
-Line 111: suggest change “bond” to “association” as bond suggest something more chemically related
-Line 112: extra comma at the end
-Line 118: typo “through”
-Line 119: “ending” should be “end”
-Line 131-132: Consider separating the sentence and subsequent changes. “…in mature leukocytes. This makes it difficult to determine…”
-Line 133: singular or plural?
-Line 134-135: This sentence has multiple grammatical errors. Please correct
-Line 140: comma after “leukocytes”
-Line 141: wrong capitalization
-Line 198: “establishing”
-Line 205: period missing
-Line 210: missing comma
-Line 210: “might” should be “may”
-Figure 2: Typo for “Geminal”
-Figure 2 legend: The “red color” should be “green color”?
-Line 223: “characterize” should present tense
-Line 231: “despite that”
-Line 232: DYSKERIN should be DKC (current official symbol)
-Line 246: “revealed”
-Line 248: “telomerase”
-Line 250: missing comma
-Line 255: “et al.”
-Line 257: missing comma
-Line 264: missing period
-Line 266: is it supposed to be “postponed”?
-Line 269: why is et al here is italicized? Please standardize the rest of abbreviations.
-Line 278: suggest change to “ACD is mutated with G223V”
-Line 282: “as well as binding directly”
-Line 291: Split to two sentences
-Line 299-300: Suggest change to “This appeared to be due to different canonical pathways affected between AA and EA”
-Line 308: “…aiming at more effective…”
-Line 330: Split to two sentences
-Line 343-349: Run-on sentence; please split to multiple sentences
Author Response
Dear Reviewer,
The manuscript “The relevance of telomerase in B lymphoblastic leukemia” submitted to Genes is a fairly comprehensive paper with lots of information. However, frequent typos, grammatical errors and the lack of clear organization makes it difficult read through properly. There are also lots of inconsistencies in acronym usage (e.g. B-ALL vs ALL). There are also mentions of shelterin and other accessory proteins, but they aren’t technically part of telomerase. If they are to be mentioned, perhaps the title should be expanded to “relevance of telomere and telomerase in B lymphoblastic leukemia.” Some different sections also have different acronyms and style and redundancy of information, which makes it feel like they are written by different people but not integrated well together enough.
We appreciate the comments and suggestions. A new version of the manuscript has been written and we believe that all mentioned points were fixed, including an update in the title.
When introducing telomerase functions, there should be an explanation on the distinction between canonical and noncanonical functions. The only mentions of “non-canonical” are in the abstract and conclusion. This leaves the reader guessing whether they are reading about canonical or non-canonical functions. I suggest separating them into different sections: one talking about telomere length regulation in normal blood cells and B-ALL aspects and one about gene-level regulation of TERT on other signaling pathways. It is sometimes hard to differentiate the two, and in some cases, both might be involved at the same time. That is also fair and should be mentioned as a possibility and an important consideration when discussing etiology of the disease.
We totally agree with the reviewer on the need to highlight the non-canonical functions of telomerase, but separating them into a section would lead to a redundant text, since, as the reviewer said, it is not easy to differentiate canonical and non-canonical functions regarding to cell cycle control. So, we updated the section “Telomerase and cancer” to present the non-canonical functions of telomerase in a more detailed way and reviewed the approach of this point through the text.
Another important point is that the Shelterin complex is defined set of proteins associated with the telomere. In some parts, it seems to be inferred that shelterin includes all the other protein partners related to telomerase that are not directly in the holoenzyme, which is incorrect. For subsection headers, perhaps the title of “Shelterin and other telomere-related partners” could be used.
The reviewer is right, we removed the mention of other proteins from that topic instead of changing the title, we believe it keeps the focus of the article and make the reading more straightforward.
Overall, I do like some of the information included (e.g. Table 1 and supplementary table 1) and there are definitely a lot of information in this manuscript. The tables actually have pretty nice information that is not even reflected in the text! The text itself is written in a way that is not efficient to read for readers. Please refer to the specific points below on some key issues identified. As can be seen from the list, there are also very many editorial issues that needs fixing (and likely more that I have not listed). Careful editing of the manuscript prior to next review would be greatly appreciated. Despite the organization/flow and editorial issues, I think this manuscript does have its merit but needs some major revisions.
We are grateful to the review for the comments and suggestions. In fact, a lot of information were summarized into tables in order to avoid a long and boring text, it is a strategy to show important data in a straightforward way. Some of this information is now better discussed in the new version anyway.
Specific points:
Line 83-84: It is not immediately obvious how antisense of these genes affects telomerase activity. Might be more appropriate in a later section focusing on telomerase? Or some more details might be needed.
The sentence was rewritten and appropriately replaced.
-Line 87: “with” needs to be removed. Otherwise, it reads like the quantification of these outcomes was to be included
Adjustment is done
-Line 105: Do not italicize MLL as it is not an official gene symbol under HGNC. The official gene symbol as you have mentioned is KMT2A.
Adjustment is done
-Line 114: suggest “tandem repeat sequences”
Adjustment is done
-Line 117: please specific what instability; I think you meant “genome instability” here
Adjustment is done
-Line 119: Citation 50 is not the appropriate citation. That article is genome sequencing of TERT-positive vs ALT cancer. Given that telomere shortening is a well-accepted knowledge in the field, a review paper should suffice. Also note that the end replication problem is not the only explanation (might even be a minor one). T-loop resolution and DNA damage (like the oxidative damage mentioned right after) also contributes to shortening.
Adjustment is done
-Line 140: Why “exceptional”? Is that high or low? Since that word usually means very good. Please reconsider a more appropriate word. Perhaps you meant irregular or infrequent.
Adjustment is done
-Line 142-143: Remove “also possible to state.” Seems like low confidence statement and a slight overgeneralization of cancer. More accurate to say “Telomerase is upregulated in most cancer…” which is well known already now.
Adjustment is done
-Line 145: Suggest removing “Telomerase, a ribonucleic….ending replication obstacle.” It is repetitive so it’s unnecessary here.
Adjustment is done
-Line 147: It needs to be noted somewhere that ALT is not a telomerase-dependent mechanism.
Adjustment is done
-Line 149-151: This might be better placed in the start of the next paragraph. Here, it should be noted that TERT is the limiting factor since it is heavily regulated whereas TERC is ubiquitously expressed.
Adjustment is done
-Line 152-183: This section needs to be organized better for readability and flow (perhaps separating into to subsections or paragraphs would help). For example. 1) Canonical and non-canonical functions of TERT. 2) TERT mutation has been found in many cancers. 3) How cancer achieve immortalization through telomerase reactivation and TERT expression upregulation.
We rewrote the manuscript and hope the reviewed sequence of information had improved the reading.
-Line 162-165: Are the two really distinct mechanisms? This should be a two-step mechanism. First phase, heal shortest telomere. Second phase, critical level short telomere leads to genome instability which drives telomerase activity up.
Adjustment is done
-Line 184: Wrong spelling for Shelterin. Please correct the rest. Shelterin is not typically referred in plural form.
Adjustment is done
-Line 184-210: This Shelterin part is strangely written without much aim. It is also very short so it doesn’t really convey much of anything. Since telomerase is the focus of this paper, perhaps shelterin may not need much mention other than an introduction. I suggest including this with earlier introduction to create an introduction to telomere.
We replaced all basic information about Shelterin into telomere topic. Keeping this information is necessary since we cited works that show the prognostic potential of Shelterin complex.
-Line 228-240: Why is the CLL being discussed in the section for ALL? Furthermore, CLL has not even been introduced before. Given that the paper is focusing on ALL, perhaps CLL should be mentioned in the introduction but not be discussed any further later.
It was actually a mistake. The adjustment is done.
-Line 282: What’s TA? Please define.
Adjustment is done
-Line 281-288: I am unaware that p23 being a shelterin protein, so it is strange to be discussed under this section. P23 does associate with TERT though.
It was also a mistake, what was fixed.
-Line 311: Aren’t there four compounds in table 1?
Adjustment is done
-Telomerase inhibitors section: Are any of these in clinical trials with known outcomes? This information would be useful to know.
The known outcomes in the studies are not for leukemia, so we decided do not mention to preserve the focus of the review.
-References: There are 214 references but in text there are only up to 124 citations. Where are the rest cited? Is that combined with supplementary?
Exactly. We produced a single merged references list following instructions for authors provided by the journal.
Minor points (typos, grammatical, editorial errors):
All of them were fixed.
Round 2
Reviewer 2 Report
The revised manuscript “The relevance of telomerase and telomere-associated proteins in B acute lymphoblastic leukemia” submitted to Genes is a much improved version from the previous one and has incorporated several of the changes I have suggested in the first review. The title is more appropriate now given the content of the manuscript. Organization is also improved with better flow and readability. However, there are still a few typo/grammatical/editorial errors listed below and some suggested order changes. With some minor revisions, I think this manuscript warrants publication in Genes.
Specific points:
Line 63: “focused”
Line 101: suggest change “originate” to “produce”
Line 107-136: Suggest changing the orders by introducing telomeres, then shelterin, then white blood cells (these are listed in this order in the subheader as well). It makes more sense to introduce telomeres and shelterin sequentially before talking about telomere lengths in white blood cells.
Line 162: “in different steps of this reactivation”
Line 167: suggest change to “it is becoming clear that…”
Line 179: “in which its level is associated..”
Line 209: typo “methylation”
Line 211: typo “telomerase”
Line 213: typo “group”
Line 238: “Despite that”
Line 262: “by different strategies, such as disrupting…”
Line 270: “…which limits clinical applications”
Line 277: “…aiming at targeting the canonical function…”
Author Response
Dear Reviewer,
First of all, thank you for all contributions that significantly improved our work. We performed a new revision according to the comments and suggestions. All corrections are done.
Best regards